# Magnetically Induced Flow Focusing of Non-Magnetic Microparticles in Ferrofluids under Inclined Magnetic Fields

**DOI:** 10.3390/mi10010056

**Published:** 2019-01-15

**Authors:** Laan Luo, Yongqing He

**Affiliations:** School of Chemical Engineering, Kunming University of Science and Technology, Kunming 650500, China; lla1016@163.com

**Keywords:** flow focusing, magnetic field, microparticles, ferrofluids

## Abstract

The ability to focus biological particles into a designated position of a microchannel is vital for various biological applications. This paper reports particle focusing under vertical and inclined magnetic fields. We analyzed the effect of the angle of rotation (*θ*) of the permanent magnets and the critical Reynolds number (*Re_c_*) on the particle focusing in depth. We found that a rotation angle of 10° is preferred; a particle loop has formed when *Re* < *Re_c_* and *Re_c_* of the inclined magnetic field is larger than that of the vertical magnetic field. We also conducted experiments with polystyrene particles (10.4 μm in diameter) to prove the calculations. Experimental results show that the focusing effectiveness improved with increasing applied magnetic field strength or decreasing inlet flow rate.

## 1. Introduction

Continuous flow focusing of microparticles/cells is an essential step for the downstream counting [1] and analyses [2] in microfluidics. Since it allows focusing the samples into a narrow region near the centerline of the microchannel, there is a wide range of applications [3,4,5] for increasing the detection efficiency in flow cytometry and throughput in particle sorting, as well as for protecting samples from unwanted interactions with the channel walls, which may cause shear or surface-induced damages to the sample. An efficient flow-focusing system should be able to push particles away from the walls of the channel and align them to move along defined flow paths. Focusing particles to a narrow stream, however, is not a trivial task. Due to the laminar nature of microfluidic flows, particles suspended in a fluid medium tend to follow the fluidic streamlines unless a lateral force moves them from their original paths [6]. Hydrodynamic forces have been commonly used to manipulate streamlines to guide particles to a confined region [7], which belongs to a passive focusing technique. However, many active focusing techniques, i.e., the use of external fields, have been applied to focused particles in microfluidic devices, such as acoustic, optical, and electrical focusing techniques, which typically require high power instruments (such as lasers or high voltage power supplies) [8].

Ferrofluids have demonstrated great potential for a variety of focusing of non-magnetic micro-particles/cells in microfluidics [9]. Compared to other technologies, magnetic focusing technology, which refers to the induced motion of particles in a non-uniform magnetic field, has distinct advantages, such as low cost, less sample consumption, no heating problem, and does not require expensive external systems as an aid [10]. Magnetic focusing technology, magnetophoresis, can be either positive or negative depending on whether the particle is more or less magnetizable than the suspending medium. Magnetic particles suspended in nonmagnetic solutions experience positive magnetophoresis and are directed along the magnetic field gradient toward a magnet [11]. In contrast, the non-magnetic particles suspended in the magnetic solutions are subjected to negative magnetophoresis and pushed away from the magnet [12]. The former phenomenon has been widely used to separate and sort cells or biomolecules in microfluidic devices by selectively labeling target cells with functionalized magnetic beads such as CellSearch technology [13], which is the only one approved by the US Food and Drug Administration (FDA).

In recent years, the use of negative magnetophoresis to focus particles in microchannels has attracted the attention of many scholars. Zhou et al. [14] developed a three-dimensional numerical model to simulate the transmission of diamagnetic particles during inertial focusing and magnetic separation in the entire microchannel and found that the predicted particle trajectories are roughly consistent with experimental observations. They further proposed a method for symmetrically embedding two repulsive permanent magnets around a linear rectangular microchannel in a microfluidic device based on polydimethylsiloxane (PDMS) to study the three-dimensional magnetic focusing of polystyrene particles in a ferrofluid [15]. Zhu et al. [16] have embedded permanent magnets in a PDMS-based microfluidic chip, in which magnets can be placed very close to a planar microchannel to enhance the magnetic field and field gradients. The non-magnetic particles can be focused continuously in a paramagnetic solution.

Since the end of an inclined magnet will form a high gradient region near the channel wall, an enhanced negative magnetophoresis will push the nonmagnetic particles toward the centerline by placing two opposite magnets on both sides of the channel. This situation is ideal for the magnetically induced flow focusing operation. There are few studies [17,18,19] on the ferrohydrodynamics with inclined magnetic fields, which focus on the effect of the flow patterns, inclination angle, and saturation magnetization on the heat transfer performance. To the best of our knowledge, no research explored the flow focusing of nonmagnetic particles by using the inclined magnetic fields. When the flow rate is also small, or the magnetic field is strong, the particles will be pushed back by the magnetic force before entering the magnetic field region, forming a circulation loop. It is necessary to consider the critical flow rate and the relative magnetic field parameters to control the particles accurately.

In this paper, we study the magnetically induced focusing of non-magnetic particles in ferrofluids under inclined magnetic fields. The effective focusing of the particles is achieved under different conditions including the rotation angle and critical Reynolds number. The paper is organized as follows: We first analyzed the various forces acting on the particles and determined the equation of motion of the particles. Then, we calculated the magnetic force that the particles are subjected to when the permanent magnets are placed in parallel or tilted. Again, we calculated the critical conditions, the critical magnetic field and flow rate, and compared them with the experimental results. Finally, we dimensioned the magnetic field and flow field and analyzed the focusing effectiveness of the particles at the exit of the channel.

## 2. Theoretical Analysis

Microfluidic magnetic focusing is a research field that involves the interaction between magnetism and fluid flow on a microscale [20]. In this section, we analyzed the vital forces that may affect the particle trajectory while passing through a microfluidic device. Non-magnetic microparticles in ferrofluids, under continuous flow conditions, are subject to a variety of forces, including magnetic force, viscous drag force, gravity/buoyancy, Brownian motion, the interaction between particles and surrounding medium, and interparticle effects, shown in Figure 1a,b. We first estimate the order of magnitude of each force to identify the dominant force for useful focusing.

### 2.1. Magnetic Force

Non-magnetic microparticle experience a negative magnetophoretic force, ***F_m_***, in a ferrofluid when subjected to a non-uniform magnetic field. The force can be written as [21]
(1)$$F_{m}=-V_{p}\mu_{0}(M\cdot \nabla )H,$$
where *V_p_* is the volume of the individual microparticle; *μ_0_* is the magnetic permeability of the vacuum, and equal by 4π × 10^−7^ H/m; the effective magnetization of the ferrofluids around the microparticles ***M*** (could be determined by the classical Langevin theory) is collinear with a static magnetic field ***H*** produced by the permanent magnet.

### 2.2. Viscous Drag Force

In low Reynolds number microfluidic systems, the hydrodynamic drag force, ***F_d_***, is related to the size and velocity of the particle, and the drag coefficient. We use the classical Stokes’ formula to present the drag force:(2)$$F_{d}=3\pi \eta D_{p}(u_{p}-u_{f})C_{D},$$
where *η* is the viscosity of suspensions; *D_p_* is the diameter of the non-magnetic microparticles; ***u_p_*** and ***u_f_*** are the velocities of microparticles and suspensions, respectively. The coefficient, *C_D_*, accounts for the increased fluid resistance when the particle moves near the microfluidic channel surface [22]:
(3)$$C_{D}={\left[1-\frac{9}{16}\left(\frac{D_{p}}{D_{p}+2\Delta }\right)+\frac{1}{8}{\left(\frac{D_{p}}{D_{p}+2\Delta }\right)}^{3}-\frac{45}{256}{\left(\frac{D_{p}}{D_{p}+2\Delta }\right)}^{4}-\frac{1}{16}{\left(\frac{D_{p}}{D_{p}+2\Delta }\right)}^{5}\right]}^{-1},$$
where *Δ* defines the shortest distance between the particle surface and channel wall.

### 2.3. Gravity and Buoyancy

The gravity and buoyancy of the non-magnetic particles in ferrofluids can be written as a resultant force, ***F_n_***, as shown in the following equation:(4)$$F_{n}=\frac{\pi D_{p}^{3}}{6}\left(\rho_{p}-\rho_{f}\right)g,$$
where *ρ_p_* and *ρ_f_* are the density of the nonmagnetic particles and the magnetic fluid, respectively; ***g*** is the gravitational acceleration. For a 5 μm particle in EMG 607 ferrofluids (*ρ_p_* = 1064 kg/m^3^, *ρ_f_* = 1100 kg/m^3^, ***g*** = 9.8 m/s^2^), the calculated gravity and buoyancy are 6.98 × 10^−2^ pN and 7.20 × 10^−2^ pN, respectively, both of which are approximately two orders of magnitude smaller than magnetic and drag forces (~pN).

### 2.4. Brown Force

The phenomenon that suspended particles never stop moving irregularly is called Brownian motion. The Brownian force, ***F_B_***, can be expressed by the following formula: [23]
(5)$$F_{B}=\zeta \sqrt{\frac{6\pi k_{B}\eta TD_{p}}{\Delta t}},$$
where *k_B_* is the Boltzmann constant; *T* is the absolute temperature; *Δt* is the magnitude of the characteristic time step; the parameter, *ζ*, is the Gaussian random number with zero mean and unit variance. Brownian motion only affects the motion of the particle when the particle diameter is small enough (less than the critical diameter of the particle), or the ***F_m_*** applied to the particle is weak [12]. In order to be able to determine the particle diameter that affects the Brownian motion, Gerber [24] et al. used the following formula to calculate the critical diameter of the particle:(6)$$|F|D_{p}\leq k_{B}T,$$
where |*F*| is the resultant force of the non-magnetic particles. When the applied magnetic field is 1 T, the critical diameter of the Fe_4_O_3_ particles in water can be calculated as 40 nm according to the above formula.

Moreover, the interactions of the particle itself and with fluid usually could be neglected in a low concentration condition. Finally, only the magnetic force and drag force need to be considered in our study.

### 2.5. The Motion Equation of Particles

Figure 2 illustrates the focusing mechanism of non-magnetic particles in the magnet-microchannel system, where particles are pushed away from the high field region and concentrated along the centerline of the channel under the actions of two magnets with same opposite polarity. The motion of particles can be analyzed by considering dominant magnetic and hydrodynamic forces. Thus,
(7)$$m_{p}\frac{du_{p}}{dt}=F_{m}+F_{d}.$$

## 3. Materials and Methods

Figure 3a shows a picture of the microfluidic chip adopted in our experiment. The straight channel was fabricated with PDMS (Sylgard 184, Dow Corning, Midland, MI, USA) by soft lithography [25]. Neodymium iron boron (NdFeB) permanent magnets were embedded on both sides of the channel with the same pole. The angle of rotation of the permanent magnet and the distance between the magnet and the channel can be changed during the experiment. The spatial position of the magnet and the channel is shown in Figure 3b and their specific sizes are given in the following section.

We used a commercial water-based magnetite ferrofluid (EMG 607, Ferrotec Corp., Tokyo, Japan) in our experiments. The volume fraction of magnetite particles for this ferrofluid is 1.8%. Initial magnetic susceptibility is 0.36; saturation magnetization is 7962 A/m; viscosity is 1.3 × 10^−3^ N⋅s/m^2^. The particle solution was made by suspending 10.4 μm polystyrene particles (Bissler Corp., Tianjin, China) in ferrofluid at a volume ratio of 1:1. The dilute ferrofluid was prepared by mixing the original ferrofluid with pure water to a concentration of 0.022%. Tween 20 (Fisher Scientific Corp., Carlsbad, CA, USA) was added to the particle suspension at 0.1% by volume to minimize their aggregations and adhesions to channel walls.

As shown in Figure 3c, microparticles suspended in the diluted ferrofluid were driven through the microchannel by an infusion syringe pump (Harvard PHD 2000, Harvard Apparatus, Holliston, MA, USA). Two equal length tubes are inserted into the chip inlet and outlet, respectively, and bonded with glue. Before the experiment, the inlet tube was connected to the syringe, and the outlet tube was placed in the beaker. Particle motion was visualized using an inverted microscope (Nikon TE2000U, Tokyo, Japan) under bright-field illumination. Digital videos (at a time rate of around 12 frames per seconds) and images were recorded through a CCD camera (PixeLINK-B742F, Ottawa, ON, Canada) and post-processed using image analysis software (ImageJ; http://rsb.info.nih.gov/ij/).

## 4. Results and Discussion

### 4.1. Magnetic Field

The system considered in the model consists of a microfluidic channel and a permanent magnet, in which dimensions of the channel and the magnet are labeled, as illustrated in Figure 4. The magnetic force experienced by a particle in the microchannel under a magnetic field can be expanded according to Equation (1),
(8a)$$F_{mx}=-V_{p}\mu_{0}\left[M_{x}\frac{\partial H_{x}\left(x,y\right)}{\partial x}+M_{y}\frac{\partial H_{x}\left(x,y\right)}{\partial y}\right],$$
(8b)$$F_{my}=-V_{p}\mu_{0}\left[M_{x}\frac{\partial H_{y}\left(x,y\right)}{\partial x}+M_{y}\frac{\partial H_{y}\left(x,y\right)}{\partial y}\right].$$

We use the Langevin equation to obtain the relationship between the saturation moment of the Fe_3_O_4_ nanoparticles *M_d_* and the effective magnetization of the ferrofluids ***M***, as follows [21]:(9a)$$\frac{M}{\phi M_{d}}=coth\left(\alpha \right)-\frac{1}{\alpha },$$
(9b)$$\alpha =\frac{\pi \mu_{0}M_{d}Hd^{3}}{6k_{B}T},$$
where *d* = 10 nm is the average diameter of the Fe_3_O_4_ nanoparticles; ϕ is the particle volume concentration; *M_d_* = 4.423 × 10^5^ A/m can be calculated from the manufacturer-provided saturation magnetization of ferrofluids. Moreover, the analytical expressions for *x*- and *y*-component of the effective magnetization can be expressed as
(10a)$$M_{x}=M\frac{H_{x}}{H},$$
(10b)$$M_{y}=M\frac{H_{y}}{H}.$$

The analytical expressions for the *x*- and *y*-component of the magnetic field strength for single rectangular magnet are given by Equation (11a,b) [26].
(11a)$$H_{x}\left(x,y\right)=\frac{M_{s}}{4\pi }\left\{ln\left[\frac{{\left(x+0.5l_{m}\right)}^{2}+{\left(y-0.5w_{m}\right)}^{2}}{{\left(x+0.5l_{m}\right)}^{2}+{\left(y+0.5w_{m}\right)}^{2}}\right]-ln\left[\frac{{\left(x-0.5l_{m}\right)}^{2}+{\left(y-0.5w_{m}\right)}^{2}}{{\left(x-0.5l_{m}\right)}^{2}+{\left(y+0.5w_{m}\right)}^{2}}\right]\right\},$$
(11b)$$H_{y}\left(x,y\right)=\frac{M_{s}}{2\pi }\left\{tan^{-1}\left[\frac{w_{m}\left(x+0.5l_{m}\right)}{{\left(x+0.5l_{m}\right)}^{2}+y^{2}-0.25w_{m}{}^{2}}\right]-tan^{-1}\left[\frac{w_{m}\left(x-0.5l_{m}\right)}{{\left(x-0.5l_{m}\right)}^{2}+y^{2}-0.25w_{m}{}^{2}}\right]\right\}.$$

The expressions for the magnetic field gradients are
(12a)$$\frac{\partial H_{x}\left(x,y\right)}{\partial x}=\frac{M_{s}}{2\pi }\left[\begin{array}{c} \frac{x+0.5l_{m}}{{\left(x+0.5l_{m}\right)}^{2}+{\left(y-0.5w_{m}\right)}^{2}}-\frac{x+0.5l_{m}}{{\left(x+0.5l_{m}\right)}^{2}+{\left(y+0.5w_{m}\right)}^{2}}\\ -\frac{x-0.5l_{m}}{{\left(x-0.5l_{m}\right)}^{2}+{\left(y-0.5w_{m}\right)}^{2}}+\frac{x-0.5l_{m}}{{\left(x-0.5l_{m}\right)}^{2}+{\left(y+0.5w_{m}\right)}^{2}} \end{array}\right],$$
(12b)$$\frac{\partial H_{x}\left(x,y\right)}{\partial y}=\frac{M_{s}}{2\pi }\left[\begin{array}{c} \frac{y-0.5w_{m}}{{\left(x+0.5l_{m}\right)}^{2}+{\left(y-0.5w_{m}\right)}^{2}}-\frac{y-0.5w_{m}}{{\left(x-0.5l_{m}\right)}^{2}+{\left(y-0.5w_{m}\right)}^{2}}\\ -\frac{y+0.5w_{m}}{{\left(x+0.5l_{m}\right)}^{2}+{\left(y+0.5w_{m}\right)}^{2}}+\frac{y+0.5w_{m}}{{\left(x-0.5l_{m}\right)}^{2}+{\left(y+0.5w_{m}\right)}^{2}} \end{array}\right],$$
(12c)$$\frac{\partial H_{y}\left(x,y\right)}{\partial x}=\frac{M_{s}}{2\pi }\left[\begin{array}{c} \frac{w_{m}\left[y^{2}-{\left(x+0.5l_{m}\right)}^{2}-0.25w_{m}{}^{2}\right]}{{\left[{\left(x+0.5l_{m}\right)}^{2}+y^{2}-0.25w_{m}{}^{2}\right]}^{2}+w_{m}{}^{2}{\left(x+0.5l_{m}\right)}^{2}}\\ -\frac{w_{m}\left[y^{2}-{\left(x-0.5l_{m}\right)}^{2}-0.25w_{m}{}^{2}\right]}{{\left[{\left(x-0.5l_{m}\right)}^{2}+y^{2}-0.25w_{m}{}^{2}\right]}^{2}+w_{m}{}^{2}{\left(x-0.5l_{m}\right)}^{2}} \end{array}\right],$$
(12d)$$\frac{\partial H_{y}\left(x,y\right)}{\partial y}=\frac{M_{s}}{\pi }\left[\begin{array}{c} \frac{w_{m}y\left(x-0.5l_{m}\right)}{{\left[{\left(x-0.5l_{m}\right)}^{2}+y^{2}-0.25w_{m}{}^{2}\right]}^{2}+w_{m}{}^{2}{\left(x-0.5l_{m}\right)}^{2}}\\ -\frac{w_{m}y\left(x+w\right)}{{\left[{\left(x+0.5l_{m}\right)}^{2}+y^{2}-0.25w_{m}{}^{2}\right]}^{2}+w_{m}{}^{2}{\left(x+0.5l_{m}\right)}^{2}} \end{array}\right],$$
where *M_s_* = *Br*/*μ*_0_ is magnetization of the permanent magnet, *Br* is remanent magnetization of the permanent magnets.

The above formula is based on the origin of the permanent magnet as the coordinate origin, and the permanent magnet is placed vertically, that is, the magnetization direction is the *y*-direction, as shown in Figure 4b, the N-pole of the permanent magnet is on the upper side, and the S-pole is on the lower side. However, we choose the center at the entrance of the channel as the origin of the coordinates. Therefore, coordinate transformation (translation and rotation) is necessary. A rotation matrix is used to perform a rotation of points in the *xy*-Cartesian plane counter-clockwise through an angle, about the origin of the coordinate system. The coordinates after translation and rotation are obtained by using matrix multiplication. The coordinate translation is (*x*-*x_m_*, *y*+*y_m_*), and the rotation angle is *θ* (0° ≤ *θ* ≤ 90°) for the permanent magnet below the channel (Equation (13a)), while for the permanent magnet above the channel, the coordinate translation is (*x*-*x_m_*, *y*-*y_m_*) and the rotation angle is *π*-*θ*, Equation (13b). Note that the subscript 1 (and subscript 2) in the following matrix equations represent the correlation calculation of the permanent magnets below (and above) the channel.
(13a)$$\left[\begin{array}{c} x_{1}^{\prime }\\ y_{1}^{\prime } \end{array}\right]=\left[\begin{array}{cc} cos\theta & sin\theta \\ -sin\theta & cos\theta \end{array}\right]\left[\begin{array}{c} x-x_{m}\\ y+y_{m} \end{array}\right],$$
(13b)$$\left[\begin{array}{c} x_{2}^{\prime }\\ y_{2}^{\prime } \end{array}\right]=\left[\begin{array}{cc} cos\left(\pi -\theta \right) & sin\left(\pi -\theta \right)\\ -sin\left(\pi -\theta \right) & cos\left(\pi -\theta \right) \end{array}\right]\left[\begin{array}{c} x-x_{m}\\ y-y_{m} \end{array}\right].$$

Substituting the Equation (13a,b) into the Equation (11a,b), respectively, the magnetic field strength after the rotation of the permanent magnet can be obtained, and then the magnetic field strength in the coordinate system of Figure 4c can be obtained by conversion, see Equation (14a,b).
(14a)$$\left[\begin{array}{c} H_{x_{1},r}\left(x,y\right)\\ H_{y_{1},r}\left(x,y\right) \end{array}\right]=\left[\begin{array}{cc} cos\theta & -sin\theta \\ sin\theta & cos\theta \end{array}\right]\left[\begin{array}{c} H_{x_{1}^{\prime }}\left(x_{1}^{\prime },y_{1}^{\prime }\right)\\ H_{y_{1}^{\prime }}\left(x_{1}^{\prime },y_{1}^{\prime }\right) \end{array}\right],$$
(14b)$$\left[\begin{array}{c} H_{x_{2},r}\left(x,y\right)\\ H_{y_{2},r}\left(x,y\right) \end{array}\right]=\left[\begin{array}{cc} cos\left(\pi -\theta \right) & -sin\left(\pi -\theta \right)\\ sin\left(\pi -\theta \right) & cos\left(\pi -\theta \right) \end{array}\right]\left[\begin{array}{c} H_{x_{2}^{\prime }}\left(x_{2}^{\prime },y_{2}^{\prime }\right)\\ H_{y_{2}^{\prime }}\left(x_{2}^{\prime },y_{2}^{\prime }\right) \end{array}\right].$$

In the same way, we can get the rotated magnetic field gradients in the coordinate system of Figure 4c, see Equation (15a,b).
(15a)$$\left[\begin{array}{cc} \frac{\partial H_{x_{1},r}\left(x,y\right)}{\partial x} & \frac{\partial H_{x_{1},r}\left(x,y\right)}{\partial y}\\ \frac{\partial H_{y_{1},r}\left(x,y\right)}{\partial x} & \frac{\partial H_{y_{1},r}\left(x,y\right)}{\partial y} \end{array}\right]=\left[\begin{array}{cc} cos\theta & -sin\theta \\ sin\theta & cos\theta \end{array}\right]\left[\begin{array}{cc} \frac{\partial H_{x_{1}^{\prime }}\left(x_{1}^{\prime },y_{1}^{\prime }\right)}{\partial x_{1}^{\prime }} & \frac{\partial H_{x_{1}^{\prime }}\left(x_{1}^{\prime },y_{1}^{\prime }\right)}{\partial y_{1}^{\prime }}\\ \frac{\partial H_{y_{1}^{\prime }}\left(x_{1}^{\prime },y_{1}^{\prime }\right)}{\partial x_{1}^{\prime }} & \frac{\partial H_{y_{1}^{\prime }}\left(x_{1}^{\prime },y_{1}^{\prime }\right)}{\partial y_{1}^{\prime }} \end{array}\right]\left[\begin{array}{cc} cos\theta & sin\theta \\ -sin\theta & cos\theta \end{array}\right],$$
(15b)$$\left[\begin{array}{cc} \frac{\partial H_{x_{2},r}\left(x,y\right)}{\partial x} & \frac{\partial H_{x_{2},r}\left(x,y\right)}{\partial y}\\ \frac{\partial H_{y_{2},r}\left(x,y\right)}{\partial x} & \frac{\partial H_{y_{2},r}\left(x,y\right)}{\partial y} \end{array}\right]=\left[\begin{array}{cc} cos\left(\pi -\theta \right) & -sin\left(\pi -\theta \right)\\ sin\left(\pi -\theta \right) & cos\left(\pi -\theta \right) \end{array}\right]\left[\begin{array}{cc} \frac{\partial H_{x_{2}^{\prime }}\left(x_{2}^{\prime },y_{2}^{\prime }\right)}{\partial x_{2}^{\prime }} & \frac{\partial H_{x_{2}^{\prime }}\left(x_{2}^{\prime },y_{2}^{\prime }\right)}{\partial y_{2}^{\prime }}\\ \frac{\partial H_{y_{2}^{\prime }}\left(x_{2}^{\prime },y_{2}^{\prime }\right)}{\partial x_{2}^{\prime }} & \frac{\partial H_{y_{2}^{\prime }}\left(x_{2}^{\prime },y_{2}^{\prime }\right)}{\partial y_{2}^{\prime }} \end{array}\right]\left[\begin{array}{cc} cos\left(\pi -\theta \right) & sin\left(\pi -\theta \right)\\ -sin\left(\pi -\theta \right) & cos\left(\pi -\theta \right) \end{array}\right].$$

Therefore, the magnetic force acting in the *x*- and *y*-directions on the particle can be calculated by Equation (8a,b). Then we add the magnetic forces generated by the two permanent magnets at the same position to obtain the final magnetic force in the *x*- and *y*-directions.
(16a)$$F_{mx}=F_{mx_{1}}+F_{mx_{2}},$$
(16b)$$F_{my}=F_{my_{1}}+F_{my_{2}}.$$

#### 4.1.1. The Vertical Magnet Filed

The vertical magnetic field means that two permanent magnets are placed in parallel with the same polarity, i.e., *θ* = 0. Based on the above analysis, we can calculate the magnetic field strength and the magnetic force acting on the particle at the center of the channel, i.e., *y* = 0.

Figure 5a,b shows the magnetic field strength and the magnetic force generated by two permanent magnets from the channel inlet (*x* = 0 mm) to the outlet (*x* = 50 mm) at the center of the channel (*y* = 0 mm), respectively. Although the polarities of the first and second permanent magnets are opposite, the resulting magnetic field strength in the *x*-direction is the same, with a maximum of ±73.52 kA/m on the left (right) sides and a minimum of 0 kA/m in the middle. The value of the magnetic field strength in the *y*-direction is the opposite number, with a minimum of 0 kA/m on both sides and a saddle-shaped change in the middle. Therefore, the total magnetic field strength is twice the original value for the *x*-direction and 0 kA/m for the *y*-direction, as shown in Figure 5a.

The trend of the corresponding magnetic force is also the same, that is, the values in the *x*-direction (*y*-direction) are the same (opposite), as shown in Figure 5a. The magnetic force in the *y*-direction at the center of the channel is 0 N, indicating that the particles at the center are only related to the magnetic force in the *x*-direction, and the particles on both sides of the center line will move toward the center. It can be seen from the total magnetic force curve in the *x*-direction that the particles will decelerate (maximum: −25.57 pN) and accelerate (maximum: 11.53 pN) when moving at the center of the channel, and then decelerate (maximum: −11.53 pN) and accelerate (maximum: 25.57 pN).

These facts indicate that the strength of the magnetic field of the permanent magnet is proportional to the magnetic force acting on the particles; the particles at both ends of the permanent magnet are subjected to the largest magnetic force and the smallest in the middle; the particles are subjected to a negative magnetic force when initially entering the magnetic field, which will prevent the particles from continuing to move forward, after which the particles are substantially subjected to a positive magnetic force, causing the particles to accelerate.

#### 4.1.2. The Inclined Magnet Field

The inclined magnetic field means that the two permanent magnets with the same polarity are placed at an angle *θ* (0° ≤ *θ* ≤ 90°) on both sides of the channel. Analogous to the vertical magnetic field, we calculated the magnetic field strength and magnetic force at the center of the channel when the angle changes from 10° to 50°, as shown in Figure 6. Note that Figure 6a–d shows the magnetic field strength and the magnetic force of the permanent magnet below the channel; the permanent magnet above the channel is only a sign change, which can be referred to as a vertical magnetic field, which is not shown here.

A slight change in angle will cause a large change in magnetic field strength and magnetic force. Regardless of the *x*-direction or the *y*-direction, the magnetic field strength at the center of the channel decreases on the left side of the permanent magnet and increases on the right side, as shown in Figure 6a,b. For example, the magnetic field strength in the *x*-direction of the left side is reduced from |−73.52| kA/m to |−53.63| kA/m, and the right side is increased from 73.52 kA/m to 100.74 kA/m when the angle *θ* = 10°. However, we found that there is a transition on the right side when the angle is increased to 30°; increasing the angle to 40°, the magnetic field strength in the *x*-direction turns to a negative value, and the *y*-direction decreases; and then increases the angle to 50°, the magnetic field strength in *x*- and *y*-direction continues to decrease.

The trend of magnetic force is roughly the same as the strength of the magnetic field: When *θ* = 30°, there is also a transition, as shown in Figure 6c,d. Figure 6e,f show the total magnetic field strength and the total magnetic force at the center of the channel, respectively. When the permanent magnet is rotated by a certain angle, the magnetic force on the left side decreases and the right side increases. For example, when the angle *θ* = 10°, the left magnetic force decreases from |−25.57| pN to |−13.48| pN, and the right side increases from 25.57 pN to 68.65 pN.

Therefore, when the permanent magnet rotates at a certain angle, the magnetic field strength and magnetic force will also change. This does not mean that the angle can be arbitrarily rotated, because the magnetic field strength and magnetic force have a transition when the angle is rotated to 30°, which is not conducive to particle focusing. It is appropriate to choose a rotation angle of 0° < *θ* < 30°.

We also calculated the magnetic force in the *y*-direction of the half channel from *y* = 100 μm to *y* = 400 μm (*y* = 0 at the centerline of the channel, see Figure 4c), as shown in Figure 7. The maximum magnetic forces in the *y*-direction are −4.62 pN (*y* = 100 μm), −9.27 pN (*y* = 200 μm), −13.95 pN (*y* = 300 μm), and −18.68 pN (*y* = 400 μm), respectively, for the vertical magnetic field (*θ* = 0°, *Br* = 1.2 T), as shown in Figure 7a. For the inclined magnetic field (*θ* = 10°, *Br* = 1.2 T), the maximum magnetic forces are −22.03 pN, −44.36 pN, −67.27 pN, −91.11 pN, respectively, when *y* is 100 μm, 200 μm, 300 μm, and 400 μm, respectively, as shown in Figure 7b.

These values are negative because the particles are subjected to negative magnetophoresis (permanent magnet repulsive force) and move toward the centerline of the channel. For the lower half of the channel, the curve is opposite to Figure 7, with absolute values equal. It is because of the negative magnetophoresis effect of the two permanent magnets that the particles are focused in the center of the channel. And when the permanent magnet is inclined at a certain angle, the negative magnetophoresis effect is enhanced, which is more conducive to the particles focusing.

### 4.2. Flow Field

The fluid flow pattern needs to be determined to calculate the fluid flow rate in the channel. The flow state of the fluid can be judged according to the Reynolds number calculation formula:(17)$$Re=\frac{\rho UD_{h}}{\eta }=\frac{2\rho Q}{\eta \left(w_{c}+h_{c}\right)},$$
where *U* is the average fluid velocity, $D_{h}=2w_{c}h_{c}/(w_{c}+h_{c})$ is the hydraulic diameter of the rectangular microchannel, and *Q* is the volumetric flow rate. Normally, *ρ* = 10^3^ kg/m^3^, *U* = 10^−3^ m/s, *D_h_* = 10^−3^ m, *η* = 10^−3^ N⋅s/m^2^, then *Re* = 1, which belongs to laminar flow. In the case of laminar flow, the formula for calculating the fluid velocity in the channel can be obtained [27,28].
(18)$$u_{f}=\left(\frac{\Delta p}{l_{c}}\right)\left(\frac{4h_{c}^{3}}{\eta \pi^{3}}\right)\sum_{n=0}^{\infty }\frac{{\left(-1\right)}^{n}}{{\left(2n+1\right)}^{3}}\times \left[1-\frac{cosh\frac{\left(2n+1\right)\pi y}{h_{c}}}{cosh\frac{\left(2n+1\right)\pi w_{c}}{2h_{c}}}\right]\times cosh\frac{\left(2n+1\right)\pi z}{h_{c}},$$
where *Δp* is the pressure drop of the fluid along the length of the channel; *n* is an integer. Here, we introduce the channel aspect ratio *ε* = *h_c_*/*w_c_*, and studies have shown that when 1 < *ε* < 2, the calculation error will be controlled within 1%, and if the ratio of channel aspect ratio is larger, the calculation error will also increase. For the small Reynolds number flow, in order to control the calculation error in Equation (18), only when *n* = 1 and 2, the calculation result can represent the flow velocity in the channel, and the final result is more accurate, so Equation (18) can be rewritten as follows:(19)$$u_{f}=\left(\frac{\Delta p}{l_{c}}\right)\left(\frac{4h_{c}^{3}}{\eta \pi^{3}}\right)\left\{\left[1-\frac{cosh\frac{\pi y}{h_{c}}}{cosh\frac{\pi w_{c}}{2h_{c}}}\right]\times cosh\frac{\pi z}{h_{c}}-\frac{1}{27}\left[1-\frac{cosh\frac{3\pi y}{h_{c}}}{cosh\frac{3\pi w_{c}}{2h_{c}}}\right]\times cosh\frac{3\pi z}{h_{c}}\right\},$$
where the pressure drop *Δp* is a function of the volumetric flow rate *Q* of the fluid in the channel inlet and can be expressed by the following formula [29]:(20)$$\Delta p=\frac{Ql_{c}\eta \pi^{4}}{8w_{c}h_{c}^{3}}\times {\left\{\left[1-\frac{2h_{c}}{\pi w_{c}}tanh\left(\frac{\pi w_{c}}{2h_{c}}\right)\right]+\frac{1}{81}\left[1-\frac{2h_{c}}{3\pi w_{c}}tanh\left(\frac{3\pi w_{c}}{2h_{c}}\right)\right]\right\}}^{-1}.$$

Substituting Equation (20) into Equation (19) and arranging it, the final formula for the flow velocity in the channel is:(21)$$\begin{array}{ll} u_{f}=\frac{Q\pi }{2w_{c}h_{c}}\times & {\left\{\left[1-\frac{2h_{c}}{\pi w_{c}}tanh\left(\frac{\pi w_{c}}{2h_{c}}\right)\right]+\frac{1}{81}\left[1-\frac{2h_{c}}{3\pi w_{c}}tanh\left(\frac{3\pi w_{c}}{2h_{c}}\right)\right]\right\}}^{-1}\times \\ & \left\{\left[1-\frac{cosh\frac{\pi y}{h_{c}}}{cosh\frac{\pi w_{c}}{2h_{c}}}\right]\times cos\frac{\pi z}{h_{c}}-\frac{1}{27}\left[1-\frac{cosh\frac{3\pi y}{h_{c}}}{cosh\frac{3\pi w_{c}}{2h_{c}}}\right]\times cos\frac{\pi z}{h_{c}}\right\}. \end{array}$$

In the laminar flow state, the motion equation of the particles, Equation (7), can be simplified to
(22)$$F_{m}+F_{d}=0.$$

Therefore, the flow rate of the particles can be expressed as
(23a)$$u_{px}=u_{mx}+u_{fx}=\frac{F_{mx}}{3\pi \eta D_{p}C_{D}}+u_{f},$$
(23b)$$u_{py}=u_{my}+u_{fy}=\frac{F_{my}}{3\pi \eta D_{p}C_{D}}.$$

Figure 8a,b shows the flow velocity distribution of the ferrofluids and particles, and the corresponding drag coefficient. The velocity distribution of the *xy* cross-section in the rectangular channel is parabolic, which is approximately the same as the velocity distribution of the circular channel (Figure 8a). The velocity distribution of the particles is closely related to the magnitude of the magnetic force, as shown in Figure 8b. The minimum flow rate is 4.87 × 10^−4^ m/s on the left side of the permanent magnet, and the maximum flow rate is 9.02 × 10^−4^ m/s on the right side, which is consistent with the above analysis. For the *y*-direction, the effect is to push the particles toward the center of the channel. The specific analysis is given in Section 4.4.

### 4.3. Magnetic Field and Flow Field Threshold

From Figure 5b we know that *F_mx_* has a negative peak on the left side of the permanent magnet, which has a large effect on *u_mx_* from Equation (23a). Once *u_mx_* < *u_fx_*, i.e., *u_px_* < 0, the particles will move in the negative *x*-direction, forming a loop. Therefore, in order to ensure that the particles can effectively enter the magnetic field and focus to the center of the channel, it is necessary to calculate the critical value of *u_fx_* (*Q*). From the analysis in Section 4.1, we know that 0° < *θ* < 30°, so we calculated the critical values at 0°, 10°, and 20°.

Figure 9 shows the critical values in different magnetic fields (*Br*) and different angles (*θ*). Under the same magnetic field, the critical value increases with the increase of the angle, especially when *θ* = 20°, the amplitude is larger, as shown in Figure 9a,b. For *θ* = 20°, the critical value is 499.78 μL/min (*Re_c_* = 7.05) at *Br* = 1.2 T (Figure 9a) and has reached 1021.83 μL/min (*Re_c_* = 14.41) at *Br* = 2.4 T (Figure 9b) in 36.42 mm of the channel. These values are quite large for microchannels, so we do not consider this case.

Figure 9c shows the critical value under the vertical magnetic field (*θ* = 0°). The critical value is 15.50 μL/min (*Re_c_* = 0.22) for *Br* = 1.2 T and 35.65 μL/min (*Re_c_* = 0.50) for *Br* = 2.4 T; when *Br* = 3.6 T and 4.8 T, the critical value is 55.80 μL/min (*Re_c_* = 0.79) and 75.96 μL/min (*Re_c_* = 1.07), respectively. In both cases, it is located at 9.96 mm of the channel, which means that as long as the inlet flow rate *Q* exceeds this value, the particles will smoothly enter the magnetic field instead of forming a loop. Figure 9d gives the critical value of the vertical magnetic field (*θ* = 10°). The critical value for this case is generated at 34.65 mm of the channel compared to the vertical magnetic field. For *Br* = 1.2 T and 2.4 T, the critical value is 28.74 μL/min (*Re_c_* = 0.41) and 61.83 μL/min (*Re_c_* = 0.87), respectively. The critical value is 94.93 μL/min (*Re_c_* = 1.34) and 128.03 μL/min (*Re_c_* = 1.81) when *Br* = 3.6 T and 4.8 T, respectively. These values are greater than the vertical magnetic field at the same *Br*.

These results show that both the angle of the permanent magnet and the residual magnetization of the permanent magnet influence the critical flow rate, which in turn affects the particles focusing. Although the critical flow rate is exceeded, the flow rate at the end of the channel is quite large, which requires the consideration of the effect of fluid inertia on the particles. In short, we need to consider all aspects to achieve effective control of particles/cells to meet a variety of application needs.

Figure 10 shows the pictures we obtained through experiments. In the case of a vertical magnetic field, a loop is formed when *Q* is less than the critical flow rate (*Q_c_*); and an increase in the strength of the magnetic field will cause more particles to bounce back, forming a thicker loop, as shown in Figure 10a,b. The same is true for the inclined magnetic field (Figure 10c,d), and the particles in the loop of the vertical magnetic field are less than the inclined magnetic field under the same *Br*.

The particles overcome the influence of the negative magnetic force and smoothly enter the magnetic field and focus when *Q* > *Q_c_*. We took a picture of the focusing of particles near the exit of the channel, as shown in Figure 11. The experimental results in Figure 10 and Figure 11 demonstrate that the critical flow rate determined by our theoretical analysis is correct.

### 4.4. Focusing Effectiveness

The particles can be focused when *Q* > *Q_c_* (*Re* > *Re_c_*). To facilitate a quantitative comparison of non-magnetic particle focusing in ferrofluids under different *Q*, we define a dimensionless number, focusing effectiveness, to evaluate the particle focusing performance,
(24)$$\mathrm{Focusing}\;\mathrm{effectiveness}=\frac{\mathrm{Half}\;\mathrm{width}\;\mathrm{of}\;\mathrm{the}\;\mathrm{focused}\;\mathrm{particles}\;\mathrm{stream}}{\mathrm{Half}\;\mathrm{width}\;\mathrm{of}\;\mathrm{the}\;\mathrm{channel}}=\frac{y_{e}}{y_{c}},$$
where the width of the microchannel was fixed at 500 µm in all tests of this work and the width of the focused particle stream was measured near the exit of the channel on the experimental images by ImageJ.

Figure 12a shows that an increase in flow rate results in a decrease in focusing effectiveness whether *θ* varies from 0° to 10° or *Br* varies from 1.2 T to 2.4 T. For *θ* = 0° and *Br* = 1.2 T, the minimum value of *y_e_* is 263.03 μm (*Q* = 20 μL/min, *Re* = 0.28), the corresponding effectiveness is 52.61%; the maximum value is 473.94 μm (*Q* = 80 μL/min, *Re* = 1.13), and the corresponding effectiveness is 5.21%; for *θ* = 0° and *Br* = 2.4 T, the minimum and maximum value of *y_e_* is 106.67 μm (*Q* = 40 μL/min, *Re* = 0.56) and 448.89 μm (*Q* = 100 μL/min, *Re* = 1.41), respectively, and the corresponding effectiveness is 78.67% and 10.22%, respectively, as shown in Figure 12b. For *θ* = 5° and *Br* = 1.2 T, the minimum value of *y_e_* is 228.62 μm (*Q* = 30 μL/min, *Re* = 0.42), the corresponding effectiveness is 54.28%; the maximum value is 455.59 μm (*Q* = 90 μL/min, *Re* = 1.27), and the corresponding efficiency is 8.88%; for *θ* = 5° and *Br* = 2.4 T, the minimum and maximum value of *y_e_* is 217.68 μm (*Q* = 70 μL/min, *Re* = 0.99) and 492.03 μm (*Q* = 130 μL/min, *Re* = 1.83), respectively, and the corresponding effectiveness is 56.46% and 1.60%, respectively. For *θ* = 10° and *Br* = 1.2 T, the minimum value of *y_e_* is 183.58 μm (*Q* = 30 μL/min, *Re* = 0.42), the corresponding effectiveness is 63.29%; the maximum value is 454.11 μm (*Q* = 90 μL/min, *Re* = 1.27), and the corresponding efficiency is 9.18%; for *θ* = 10° and *Br* = 2.4 T, the minimum and maximum value of *y_e_* is 179.72 μm (*Q* = 70 μL/min, *Re* = 0.99) and 473.18 μm (*Q* = 130 μL/min, *Re* = 1.83), respectively, and the corresponding effectiveness is 64.06% and 5.36%, respectively.

These facts indicate that the focusing effectiveness is better when *Q* is near *Q_c_* and the focusing effectiveness of *Br* = 1.2 T is less than *Br* = 2.4 T when *θ* is the same. Therefore, choosing the right angle and flow rate is critical to control the particles/cells accurately.

## 5. Conclusions

The focusing of biological and synthetic particles in microfluidic devices is a crucial step for the construction of many microstructured materials as well as for medical applications. We analyzed the performance of the magnetic focusing of the particles under a vertical magnetic field and an inclined magnetic field. We found that when the rotation angle of the permanent magnet exceeds 30°, an inclined magnetic field is unsuitable for particle focusing. We also calculated the effect of the flow field on the particle focusing, i.e., the critical flow rate. The calculations show that when the rotation angle exceeds 20°, the critical minimum flow rate exceeds 500 μL/min, which is quite large for the microchannel. Therefore, the optimum value of the rotation angle of the magnets should be between 0° and 20°, and 10° is preferred. The results also show that the critical flow rate is 15.5 μL/min and 35.65 μL/min for *θ* = 0°, *Br* = 1.2 T and 2.4 T, respectively; for *θ* = 10°, *Br* = 1.2 T and 2.4 T, the critical flow rate is 28.74 μL/min and 61.83 μL/min, respectively. These relationships have been verified with experimental results, which agree quantitatively with the predictions.

## Figures and Tables

**Figure 1 micromachines-10-00056-f001:**
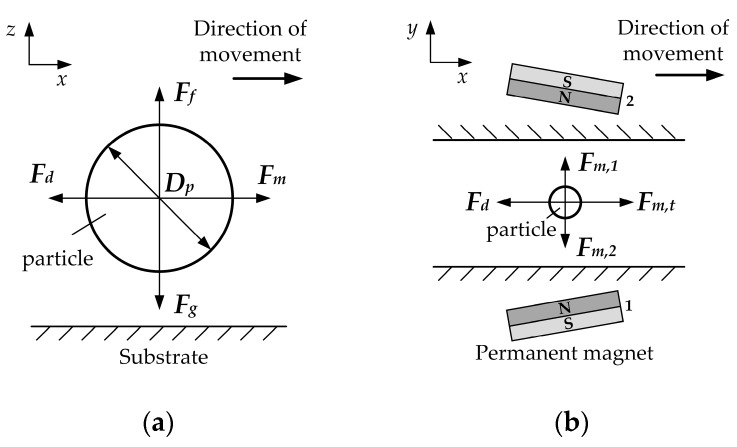
Schematic of the forces acting on the particle. (**a**) Forces analysis of a non-magnetic particle in the *x*, *z*-direction; (**b**) forces exerted on the particle when two permanent magnets are opposite in polarity.

**Figure 2 micromachines-10-00056-f002:**
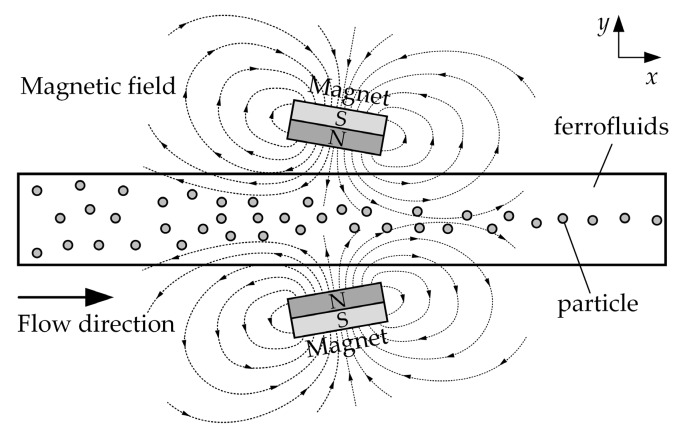
Schematic of the particles focusing mechanism. When two rectangular permanent magnets are placed at the center of a channel length with the same opposite polarity, and its direction of magnetization perpendicular to the channel wall, the magnet field magnetizes the ferrofluids within the microchannel and subsequently affects the particle motion.

**Figure 3 micromachines-10-00056-f003:**
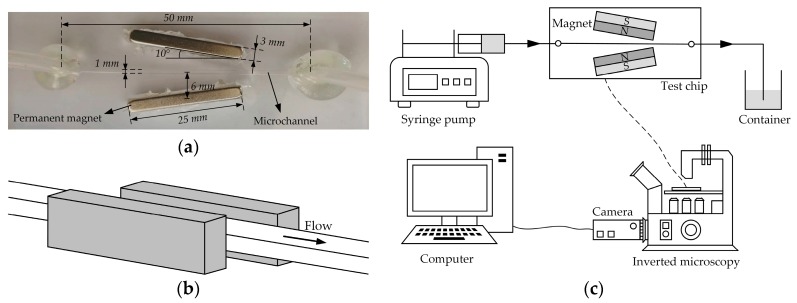
Experimental chip and system: (**a**) picture of the microfluidic chip with permanent magnets embedded in polydimethylsiloxane (PDMS); (**b**) schematic diagram of the spatial position of the channel and magnet, and note that the centerline of the permanent magnet is on the same plane as the centerline of the channel; (**c**) the experimental setup.

**Figure 4 micromachines-10-00056-f004:**
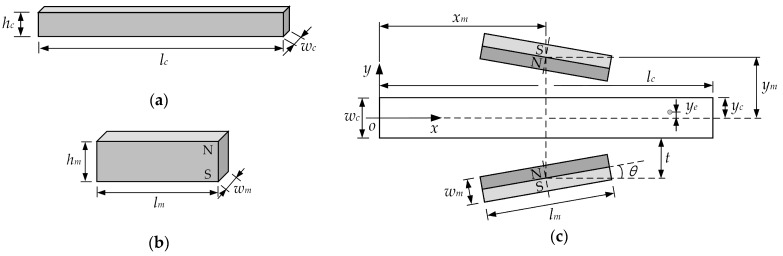
The dimensions of (**a**) the microfluidic channel and (**b**) the permanent magnets; (**c**) the relative locations of the microchannel and the magnets: *l_c_* = 50 mm, *w_c_* = 1 mm, *h_c_* = 1 mm, *l_m_* = 25 mm, *w_m_* = 3 mm, *h_m_* = 10 mm, *t* = 5.5 mm, *x_m_* = 25 mm, *y_m_* = 6 mm; *x*-*y* coordinate system is within the microchannel, with its origin at the center of the cross-section of the microchanel.

**Figure 5 micromachines-10-00056-f005:**
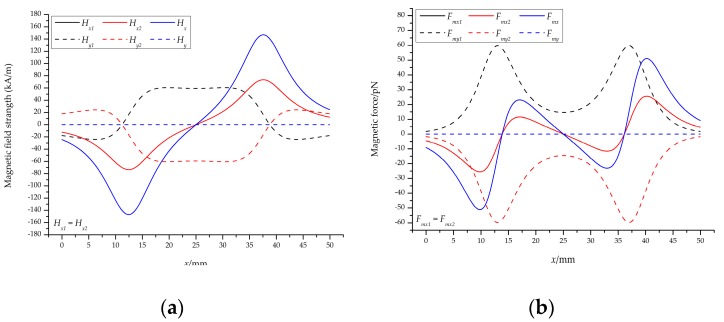
The results of vertical magnetic field: (**a**) The magnetic field strength and (**b**) the magnetic force acting on the particles at the center of the channel. Note that the residual magnetization of a single permanent magnet is 1.2 T.

**Figure 6 micromachines-10-00056-f006:**
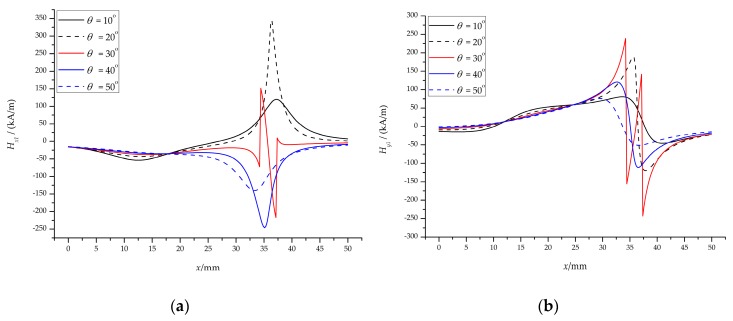
The results of the inclined magnetic field: The magnetic field strength (**a**) in the *x*-direction and (**b**) in the *y*-direction; the magnetic force (**c**) in the *x*-direction and (**d**) in the *y*-direction; (**e**) the total magnetic field strength and (**f**) the total magnetic force at the center of the channel.

**Figure 7 micromachines-10-00056-f007:**
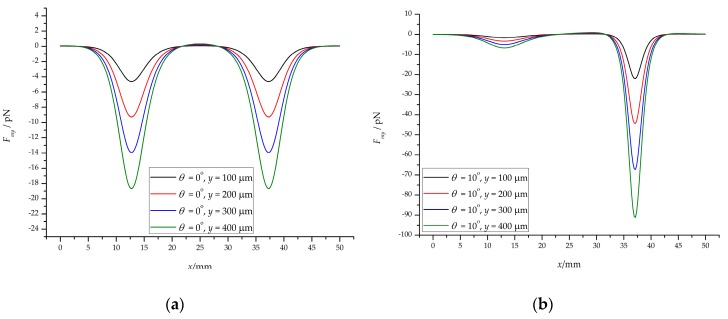
The results of magnetic force in the *y*-direction (half channel): (**a**) the vertical magnetic field; (**b**) the inclined magnetic field.

**Figure 8 micromachines-10-00056-f008:**
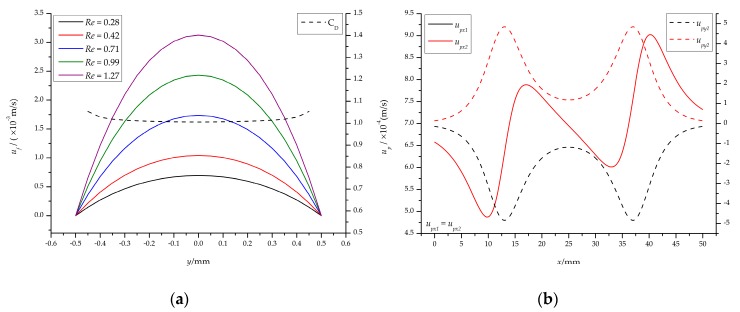
The results of flow field: (**a**) The flow rate of the ferrofluids in the channel at different inlet flow rates and drag coefficient along the *y*-direction; (**b**) the velocity distribution of particles in the center of the channel at *Q* = 20 μL/min (*Re* = 0.28).

**Figure 9 micromachines-10-00056-f009:**
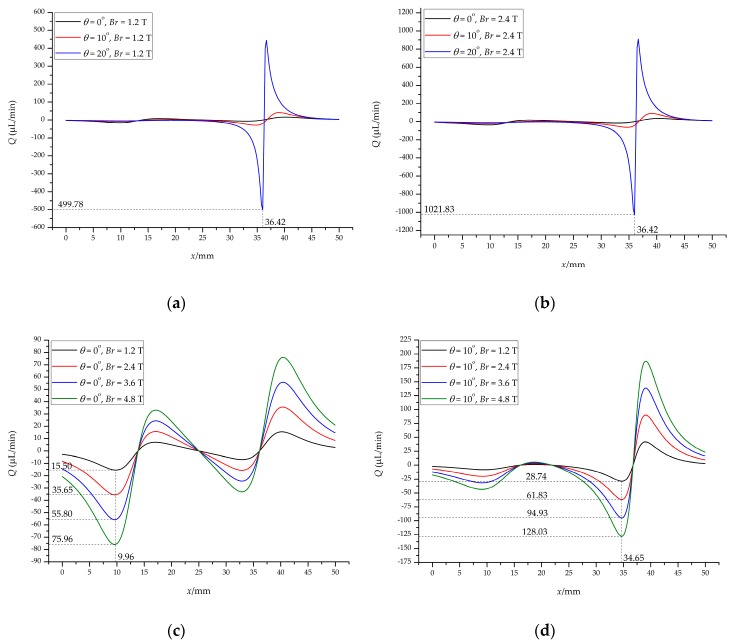
The threshold of magnetic field and flow field: the corresponding critical value of the permanent magnet after rotating at different angles for (**a**) *Br* = 1.2 T and (**b**) *Br* = 2.4 T; the corresponding critical value of (**c**) the vertical magnetic field and (**d**) the inclined magnetic field under different *Br*.

**Figure 10 micromachines-10-00056-f010:**
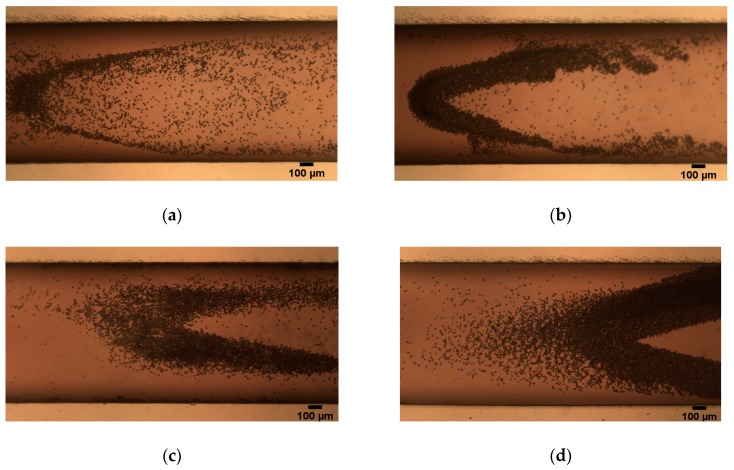
A loop formed below the critical flow rate. The vertical magnetic field (*θ* = 0°): (**a**) *Br* = 1.2 T, *Q* = 10 μL/min (*Re* = 0.01) and (**b**) *Br* = 2.4 T, *Q* = 30 μL/min (*Re* = 0.42); the inclined magnetic field (*θ* = 10°): (**c**) *Br* = 1.2 T, *Q* = 20 μL/min (*Re* = 0.28) and (**d**) *Br* = 2.4 T, *Q* = 50 μL/min (*Re* = 0.71).

**Figure 11 micromachines-10-00056-f011:**
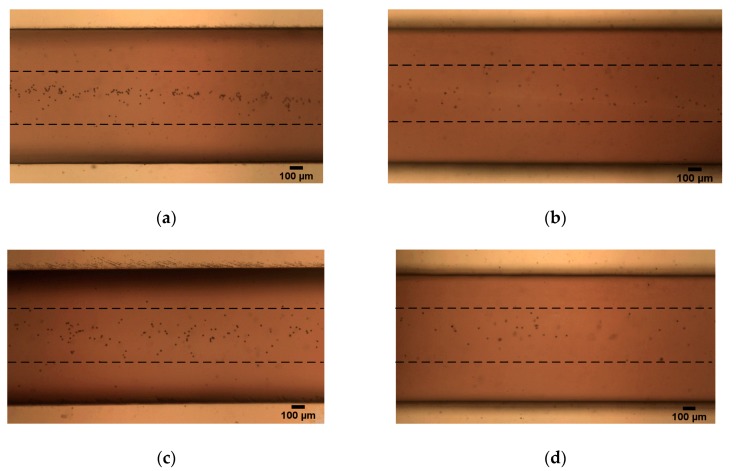
Focusing near the exit of the channel under the vertical magnetic field (*θ* = 0°): (**a**) *Br* = 1.2 T, *Q* = 20 μL/min (*Re* = 0.28) and (**b**) *Br* = 2.4 T, *Q* = 40 μL/min (*Re* = 0.56); the inclined magnetic field (*θ* = 10°): (**c**) *Br* = 1.2 T, *Q* = 30 μL/min (*Re* = 0.42) and (**d**) *Br* = 2.4 T, *Q* = 70 μL/min (*Re* = 0.99).

**Figure 12 micromachines-10-00056-f012:**
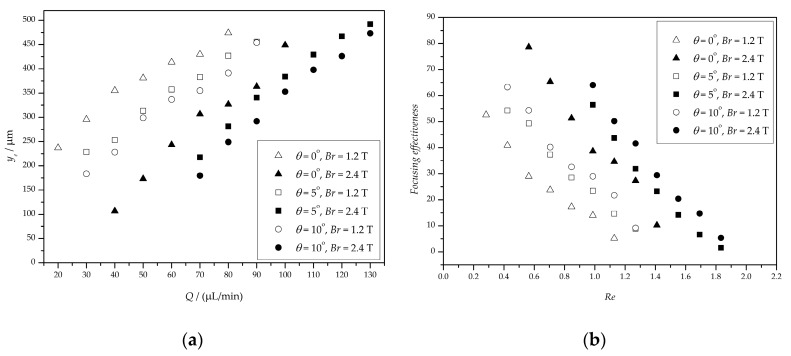
The results of focusing particles at different flow rates. (**a**) The width of the focused particle stream; (**b**) the focusing effectiveness.

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
