# Peer review of "Magnetically Induced Flow Focusing of Non-Magnetic Microparticles in Ferrofluids under Inclined Magnetic Fields"

_micromachines, 2019, doi:10.3390/mi10010056_

Round 1

Reviewer 1 Report

In this paper, the authors developed ferrohydrodynamic focusing of the non-magnetic particles under the tilted magnets in a straight channel. They analyzed the effect of the angle of rotation of the permanent magnets, the critical flow rate, and focusing effectiveness. The model was validated using analytical solution and experiments. The authors also investigated the effective rotation angle of permanent magnet. I recommend the manuscript for publication in Micromachines after minor revision.

Comments:

The authors show the vertical and inclined magnetic field and forces along the centerline of the channel (Fig. 5 and 6). It is impressive that the magnetic force changes greatly even if the rotation angle of the permanent magnet changes slightly. However, the force in the y direction of the entire channel rather than the centerline is more dominant in order to focus the particle. So I think it would be better to add contours for magnetic force of the entire (half) channel according to the some rotation angles.

In section 4.1 the authors conclude that the appropriate angle for particle focusing is between 0 and 30 degrees. The, in section 4.3, the critical values for 0, 10, and 20 degrees are obtained. I think the number of data seems to be too small. It would be better to show the result from more diverse angles in the appropriate ranges.

Minor comments:

- It would be helpful to include in the introduction some relevant papers that consider ferro hydro dynamics with two inclined permanent magnets: Journal of Magnetism and Magnetic Materials 472(2019) 115-122.

- It would be better to add a description of n in Eq. 18 on page 10.

Author Response

Please find the response in the attachment, thanks.

Reviewer 2 Report

Review report on: Magnetically induced flow focusing of non-magnetic microparticles in a straight channel

The article describes the motion of microparticles in a microchannel under the influence of a magnetic field. The authors derive the equations governing the motion of the particles under an “inclined” magnetic field. The theory developed is consistent and is presented in an easy-to-understand manner. While the manuscript investigates a topic with clear significance, I have the following suggestions to improve the article:

1.      In my view, the title misses out on the information that a ferrofluid is required for this sort of study. Perhaps, it would be appropriate to include that in the title since the entire focusing strategy relies on the presence of a ferrofluid.

2.      On a related note, I believe that the authors should discuss this point either in the Introduction or in the Conclusions. The requirement of ferrofluid is a significant limitation on the capability of this technology since different applications may not allow the use of ferrofluids. Other techniques such as acoustic –based focusing can overcome these, albeit they have their own limitations.

3.      I felt that the presentation of the theoretical analysis can be improved in terms of the post-analysis. More specifically, after arriving at Eq. 16, it would be nice to clearly delineate all the parameters that affect particle motion and clearly show how each of these parameters (or variables) influences the particle motion. It would be nice to present particle trajectories for a range of such variables. Currently, I feel that the authors focus a bit more on the specific quantitative aspects (and associated values) and less on the big take-away messages for the reader.

4.      On a related note, the authors should also comment on the limit of applicability of their analysis. For instance, the critical angle and flow rate values are obtained for the specific case considered in the manuscript, but as mentioned above, it would be nice for the reader to understand how these values would change for different cases.

Overall, I feel the article has obvious merits in that it presents a theoretical analysis of particle motion in microchannels under the influence of magnetic field. However, the presentation of the article can be improved in the above-mentioned manners to accentuate the significance of the study for a general reader.

Author Response

(The authors gave the same response as above.)

Reviewer 3 Report

In this work, authors investigated the magnetically-induced focusing of non-magnetic particles in straight channel. By using various rotation angles during analysis, the optimum rotation angle of the magnets were concluded to be between 0° and 20° (10° is preferred). The conclusions and results are well supported by the analysis of the conducted experiments and presented data but also suffer some problems. Therefore, this manuscript can be accepted for publication after some major revisions as suggested below.

1.     For Fig. 5, 7 and8, it would be nice to use different color or curves with different data markers to illustrate different curves in the same figure. Most of the curves in these figures in the manuscript are hard to read.

2.     Authors mentioned the size of the particles used is 10.4 ÎĽm, which is considered to be a relatively large size in the application of bioparticle (such as virus or protein) detection. I am curious on how the size of particles used is going to affect the entire flow speed.

3.     In Fig 11, are these data points obtained through one-time tests? In addition, some more discussion about the figure would be necessary instead of just list our the data demonstrated on graph.

Author Response

(The authors gave the same response as above.)
